# Determinants of Continued Breastfeeding at 12 and 24 Months: Results of an Australian Cohort Study

**DOI:** 10.3390/ijerph16203980

**Published:** 2019-10-18

**Authors:** Jane Scott, Ellen Ahwong, Gemma Devenish, Diep Ha, Loc Do

**Affiliations:** 1School of Public Health, Curtin University, Perth, WA 6102, Australia; ellen.ahwong@health.wa.gov.au (E.A.); gemma.devenish@curtin.edu.au (G.D.); 2Australian Research Centre for Population Oral Health, The University of Adelaide, Adelaide, SA 5000, Australia; diep.ha@adelaide.edu.au (D.H.); loc.do@adelaide.edu.au (L.D.)

**Keywords:** continued breastfeeding, determinants, formula, sociodemographic

## Abstract

Breastfeeding to 12 months and beyond offers considerable health benefits to both infants and mothers. Despite these recognized benefits, relatively few women in high income countries breastfeed for 12 months, and rarely breastfeed to 24 months. The aim of this study was to identify the prevalence and determinants of continued breastfeeding to 12 and 24 months amongst a cohort of Australian women participating in the Adelaide-based Study of Mothers’ and Infants’ Life Events affecting oral health (SMILE). Duration of breastfeeding was known for 1450 participants and was derived from feeding related data collected at birth, 3, 6, 12 and 24 months. Multivariable logistic regression analysis was used to investigate the relationship between explanatory variables and continued breastfeeding to 12 and 24 months. In total, 31.8% of women breastfed to 12 months and 7.5% to 24 months. Women who were multiparous, university educated, had not returned to work by 12 months and whose partners preferred breastfeeding over bottle feeding were more likely to be breastfeeding at 12 months. While women who had introduced complementary foods before 17 weeks and formula at any age were less likely to be breastfeeding at 12 months. Mothers who were born in Asian countries other than India and China, had not returned to work by 12 months and had not introduced formula were more likely to be breastfeeding at 24 months. The majority of the determinants of continued breastfeeding are either modifiable or could be used to identify women who would benefit from additional breastfeeding support and encouragement.

## 1. Introduction

The World Health Organization recommends that infants be exclusively breastfed for the first six months of life, after which time nutritionally adequate complementary foods should be introduced and breastfeeding continued to at least 2 years of age [1]. The importance of breastfeeding for the growth and health of infants in the first year of life is well-established and there is convincing evidence that breastfeeding beyond 12 months has a positive influence on a child’s health and development [2,3]. In addition to being a source of nutrients, breast milk includes a host of bioactive components that guide the development of an infant’s immune system [4]. Research has shown that these immunological factors are maintained to two years [5] and protection against mortality and morbidity from infectious diseases extends well into the second year of life [3]. Women who breastfeed for 12 months and beyond have been found to be more in tune with their infant’s satiety and hunger cues, enabling the establishment of better eating patterns and likely reducing the risk of obesity [6,7]. A clear inverse dose response relationship between breastfeeding duration and risk of obesity has been established [8] and in a recent study, an Australian cohort of children breastfed for 52 weeks had half the risk of being overweight or obese at 24−36 months compared with those never breastfed or breastfed for less than 17 weeks [9]. 

Research has shown that breastfeeding can also benefit the mother by reducing the risk of breast cancer [10], ovarian cancer [11], type 2 diabetes mellitus [12], hypertension [13], metabolic syndrome [14], cardiovascular disease [15,16] and possibly osteoporosis [17]. In all of these studies, an inverse association between each outcome was seen with longer duration per child and/or cumulative (lifetime) duration of breastfeeding. 

While the World Health Organization (WHO) recommends that women breastfeed their infants to 2 years and beyond [1], and despite the recognized benefits for infants and mothers, few Australian women contemplate breastfeeding to this age [18]. For this reason, breastfeeding recommendations issued by relevant authorities in Australia [19] and other high income countries such as the USA [20] have modified the WHO guidelines and recommend a culturally more attainable goal that women breastfeed to 12 months and beyond. Even so, relatively few women in most high income countries achieve this recommendation, where on average the prevalence is lower than 20% [3]. In Australia, the most recent data on breastfeeding practices come from the 2014–15 National Health Survey which reported that 27.5% of children aged 13−24 months were breastfed at 12 months [21]. 

Several terms have been used to describe breastfeeding beyond the first year of life including ‘continued’ [3], ‘sustained’ [2], ‘extended’, ‘long-term’ and ‘prolonged’ [22]. On their own, these terms are culturally subjective unless anchored with an age. Furthermore, several of these terms, the latter two in particular, may have negative connotations in high income countries, where disapproving attitudes increase, and support for breastfeeding decreases, as the age of the child increases [22,23,24]. Care therefore, needs to be taken in the use of terminology when encouraging women to continue to breastfeed beyond 12 months. For the purposes of this paper we have chosen the term ‘continued breastfeeding’ as used by Victora et al. [3] in the recent Lancet Breastfeeding Series. The aim of this study was to identify the prevalence and determinants of continued breastfeeding to 12 and 24 months amongst a cohort of Australian women.

## 2. Materials and Methods

The study is a secondary analysis of infant feeding data collected as part of the Study of Mothers’ and Infants’ Life Events Affecting Oral Health (SMILE), a population-based longitudinal birth cohort study [25]. SMILE was originally funded to recruit and follow a cohort of socioeconomically-diverse South Australian newborns from birth into their third year of life, and additional funding has been secured to follow the SMILE cohort until 7 years of age. The primary health outcomes of SMILE are two related conditions: early childhood dental caries and obesity/overweight.

### 2.1. Setting and Recruitment 

Between July 2013 until August 2014 a total of 2147 women and 2181 newborns, including 34 pairs of twins, were recruited from three major maternity hospitals in Adelaide, Australia. All new mothers with sufficient competency in English to understand the description and instructions of the study were invited to participate. While there were no specific exclusions related to gestational age and birthweight, mothers with infants in the NICU were not invited to participate. Those women who indicated their intention to move out of the greater Adelaide area within a year were excluded. Efforts were made to over-recruit from those hospitals which service the more socially disadvantaged communities in Adelaide to account for an anticipated higher attrition rate.

Women were recruited, from the postnatal wards of the participating hospitals, usually within 48 h of giving birth. Those agreeing to participate in the study were invited to complete a baseline questionnaire and follow-up questionnaires at 3, 6, 12 and 24 months. Further details of the study protocol have been reported elsewhere [25].

### 2.2. Statistical Analysis

Of the 2147 women recruited, 2112 (98.3%) completed a baseline questionnaire. The analysis population consists of the 1450 women for whom breastfeeding duration was known, that is they were known to have stopped breastfeeding prior to, or to still be breastfeeding at, 24 months. The primary outcome variables in this analysis are continued breastfeeding at 12 and 24 months postpartum. These variables were derived from questions on infant feeding practices—including current feeding method, age of cessation of breastfeeding and age of introduction of formula, complementary foods and other beverages—collected from follow-up questionnaires (either postal or online), self-completed by women at baseline and when their child was 3, 6, 12 and 24 months of age. With regards to breastfeeding practices, no distinction was made between whether a mother was directly breastfeeding from the breast or feeding an infant expressed breast milk via a bottle. The age at which these feeding events occurred were reported in weeks and/or months. Ages reported in weeks were divided by 4.33 and rounded to calculate age in months, with 26, 52 and 104 weeks representing 6, 12 and 24 months respectively. 

Explanatory variables known or suspected to be associated with maintenance of breastfeeding to 12 months or more [26] were derived from the baseline questionnaire and included: mother’s age in years (<25, 25−34 or ≥35 years); education (high school/vocational or some/completed university); country of birth (Australia/New Zealand, UK, India, China, Asia-other or Other); age of child when mother returned to work (before 12 months or not by 12 months); parity (primiparous or multiparous); pre-pregnancy body mass index (BMI) (<25, ≥25 kg/m^2^) based on self-reported weight and height; partner’s feeding preference (prefers breastfeeding or prefers formula /ambivalent) as perceived by the mother. The Index of Relative Socio-Economic Advantage and Disadvantage (IRSAD) was generated using household postcode and used as a measure of socioeconomic position (SEP)[27]. The IRSAD was categorized from deciles into 5 groups from most to least disadvantaged. Infant related explanatory variables included infant sex; birth weight (<2500, 2500–3999 or 4000 g); age of introduction of solids (<17, 17−25 or ≥26 weeks) and age when formula was introduced (<4 weeks, 4–7 weeks, 8−15 weeks, 16−25 weeks, ≥26 weeks or never received formula).

Simple binary logistic regression was used to investigate the relationship between explanatory variables and continued breastfeeding to 12 and 24 months (results not shown). To reduce the risk of residual confounding those variables in the simple analysis with a *p*-value < 0.20 were entered into a multivariable model [28]. Results are reported as adjusted odds ratios (AOR) with 95% confidence intervals (CI). 

There were minimal missing data for sociodemographic variables. No attempt was made to impute missing data for these variables on the basis that a previous sensitivity analysis for other dietary outcomes investigated in this cohort, in which missing data for sociodemographic explanatory variables were imputed under the assumption that data were missing at random, revealed that distributions of variables in the imputed data sets were consistent with the complete case data [29]. However, data on age at which the mother returned to work were missing for a substantial number of cases (19.2%), and a sensitivity analysis was conducted with the multivariable analyses being repeated without this variable to determine if its exclusion changed the findings of this study. 

The study was approved by the Southern Adelaide Clinical Human Research Ethics Committee (HREC/50.13, approval date: 28 Feb 2013) and the South Australian Women and Children Health Network (HREC/13/WCHN/69, approval date: 7 Aug 2013). Signed informed consent was obtained from women who were advised that their participation was voluntary and that they could withdraw at any time without prejudice. 

## 3. Results

### 3.1. Subject Characteristics

The majority of women in this analysis (n = 1431) were aged 25 to 34 years (66.1%), had commenced or completed university (50.8%) and were born in Australia (71.4%) (Table 1). Mothers who provided baseline data but without data on breastfeeding status at 24 months were younger, less educated, more socially disadvantaged and more likely to have been born outside of Australia. The population was similar to the profile reported by the Pregnancy Outcome Unit for South Australia for births in 2013 with regards to country of birth, but our sample consisted of slightly fewer younger (<25 years) mothers [30] (Appendix A). 

Almost all women (94.9%) had initiated breastfeeding and 31.8% had breastfed to 12 months and 7.5% to 24 months (Table 2 and Appendix A).

The overall median duration of breastfeeding was 28.7 weeks and, with the exception of women born in the United Kingdom (UK) and Ireland, the median duration for women born in Australia (26.5 weeks) was significantly lower (*p* < 0.001) than for women born in other countries (Table 3).

### 3.2. Determinants of Breastfeeding to 12 Months

Women who were multiparous (AOR 1.52, 95%CI 1.08−2.15), had commenced or completed a university degree (AOR 2.28, 95%CI 1.57−3.31) and had not returned to work within 12 months of the birth of their child (AOR 1.45, 95% CI 1.04–2.02) were significantly more likely to breastfeed to 12 months and beyond compared with primiparous women, those with a high school or vocational level education and those who had returned to work, respectively (Table 4). Women whose partners preferred breastfeeding were 76% more likely to breastfeed to 12 months than those who reported that their partner preferred formula feeding, or was ambivalent about how they fed their child (AOR 1.76, 95%CI 1.22−2.56). Finally breastfeeding at 12 months was independently negatively associated with the use of formula and the very early introduction of complementary foods. Women who introduced complementary foods before 17 weeks were less likely to be breastfeeding at 12 months (AOR 0.43, 95%CI 0.23−0.80) compared women who introduced complementary foods at or after 26 weeks. The introduction of formula at any age before 12 months was strongly negatively associated with breastfeeding at 12 months. There was no independent association with maternal age or country of birth.

### 3.3. Determinants of Breastfeeding to 24 Months

Again the use of formula at any age up to 12 months was strongly negatively associated with the odds of breastfeeding to 24 months, and women who had not returned to work by 12 months were significantly more likely to breastfeed to 24 months than those who had returned to work (AOR 2.58, 95%CI 1.56−4.31). While mother’s country of birth was not independently associated with breastfeeding to 12 months, women born in Asian countries other than India and China (AOR 2.88, 95%CI 1.35−6.11) had significantly greater odds of breastfeeding to 24 months compared with Australian women.

### 3.4. Sensitivity Analysis

The exclusion of the explanatory variable ‘age of infant when mother returned to work’ from the breastfeeding at 12 months model resulted in maternal age being a significant independent predictor of breastfeeding, with younger mothers (<25 years) being less likely to breastfeed to 12 months compared to older mothers (≥35 years); all other variables remained significant (Appendix A). However, the exclusion of this variable from the breastfeeding at 24 months model had no effect on the results with age of introduction of formula and maternal country of birth remaining as significant predictors of breastfeeding to 24 months.

## 4. Discussion

This research provides insight into the prevalence of continued breastfeeding to 12 and 24 months amongst a contemporary cohort of South Australian mothers. In this study, just under one third of women continued breastfeeding to 12 months or beyond which is similar to that reported for the USA (30.7%) [31] and Norway (36%) [32]. When compared to earlier Australian studies, including a secondary analysis of the 2010 Australian National Infant Feeding Survey (31.2%) [33], the 2014–15 National Health Survey (27.5%) [21] and the 2004 national Longitudinal Study of Australian Children (LSAC) (30%) [34], the proportion of infants breastfed to 12 months has remained relatively unchanged over the last decade or so. Hence, the majority of Australian infants and their mothers are continuing to be deprived of the considerable benefits of continued breastfeeding.

Compared to low income countries where more than 60% of children are breastfed for 20 to 23 months [3], breastfeeding to 2 years is rare in Australia and other high income countries. In this study, only 7.5% of mothers were still breastfeeding at 2 years which was similar to the 5.6% of women reported to breastfeed beyond 24 months in a large Canadian study [35]. Data from the 2011 UK Diet and Nutrition Survey of Infants and Young Children suggests that the prevalence would be even lower in the UK as only 8% of children aged 12 to18 months were still being breastfed [36]. Slightly higher prevalence rates have been reported for a study of women in northern Italy (12%) [37] and a study of WIC participants in California (11%) [38].

This study identified a significant association between continued breastfeeding and a number of sociodemographic factors, although not all factors were associated with continued breastfeeding to both 12 and 24 months. Consistent with other studies of women from high income countries [26], level of maternal education was strongly associated with continued breastfeeding and compared to women with a high school or vocational education, those with at least some university education were more than twice as likely to breastfeed to 12 months. Secondary analysis of the 2010 NIFS data found also that university educated women were more likely to breastfeed to 12 months than less educated women [33], and this association between continued breastfeeding to 12 months and higher level of education has been reported in the Australian Longitudinal Study on Women’s Health (ALSWH) [39] as well as US [40] and Italian [37] studies.

Parity was independently associated with continued breastfeeding, with the odds of breastfeeding to 12 months being 52% higher for multiparous women compared to first time mothers. Parity has not been consistently associated with continued breastfeeding [26] however, in Australia where at least nine out of every 10 women initiate breastfeeding [41], multiparity may serve as a proxy measure of prior breastfeeding experience which has been associated with continued breastfeeding [26]. For example, a longitudinal analysis of the ALSWH revealed that Australian women were much more likely to breastfeed their second child for 6 months or more if they had breastfed their first child for at least 6 months [39]. This finding is important as it highlights the importance of helping first time mothers to successfully establish breastfeeding and achieve their intended duration goals so that subsequent infants can benefit from this experience. It also highlights the importance of identifying multiparous women who have had an unsuccessful breastfeeding experience with their first infant and providing them with additional support to increase the duration of breastfeeding for the latest and subsequent children [39].

While foreign born mothers have been shown to be more likely to breastfeed to 12 months than US-born mothers [38] and Swedish-born mothers [42], the association between country of birth and continued breastfeeding has been rarely investigated in Australian studies. The median duration of breastfeeding in this study was significantly lower amongst Australian and UK-born women than women born in India, China and other Asian countries. While maternal country of birth was not associated with continued breastfeeding to 12 months, women born in Asian countries other than India and China were more likely to breastfeed to 24 months than Australian women. Women in this latter group were predominantly from South East Asian countries such as the Philippines, Thailand, and Vietnam which are known for their high rates of continued breastfeeding [3]. There is evidence that migrant mothers become acculturated and adopt the breastfeeding practices of their host country [43,44] and therefore, women from countries with a tradition of continued breastfeeding should be encouraged and supported to maintain their cultural practices in Australia.

Compared to women who entered pregnancy with a healthy BMI, those with a high pre-pregnancy BMI were roughly half as likely to continue breastfeeding to 12 months. Excess body weight has been consistently associated with a lower odds of initiating breastfeeding and a shorter duration of exclusive and any breastfeeding [45,46]. Women with obesity are more likely to deliver by caesarean section [47] and therefore are more likely to miss out on the opportunity to practice early skin-to-skin feeding [48], which in turn has been shown to be more closely associated with exclusive breastfeeding among mothers with obesity than other mothers [48]. Physical barriers such as larger breasts, bigger areolas and additional body tissue can make positioning and attachment difficult [49], while having overweight and obesity have been associated with delayed onset of lactation [46]. Embarrassment and discomfort with breastfeeding is heightened in women with overweight and obesity, even in the hospital setting where women may be in a shared room or open ward with a constant stream of familiar and unfamiliar visitors [49]. Women with overweight and obesity require additional support and guidance in hospital to overcome the physical barriers associated with larger breasts, and strategies and support from partners and family members to tackle the perceived stigma associated with obesity and breastfeeding in public [49].

The importance of partner support for breastfeeding was demonstrated in this study, with women whose partners preferred breastfeeding being 76% more likely to breastfeed to 12 months than those whose partners either preferred formula feeding or were ambivalent about the feeding method. Partner approval and support of breastfeeding has been consistently shown to be a key determinant of breastfeeding success including the decision to breastfeed and to continue breastfeeding [50,51]. Partner support is likely to be even more important in the case of continued breastfeeding to 12 months and beyond, as societal acceptance, particularly amongst men, of continued breastfeeding and breastfeeding in public decreases as the age of the breastfed child increases [23,24].

Continued breastfeeding was negatively associated with maternal employment, with those who had not returned to work by the time their child was 12 months old being significantly more likely to be breastfeeding at 12 and 24 months. Maternal employment has not consistently been associated with continued breastfeeding to 12 months, with a recent review reporting that most studies that assessed maternal work failed to find an association [26]. However, a study of WIC participants in California reported that women who returned to work within 3 months of delivery were significantly less likely to breastfeed to either 12 or 24 months postpartum compared to women who returned to work after their infant was 7 months or older [38]. Similarly, participation in childcare, which can be taken as a proxy indicator of maternal employment, has been shown to be associated with a reduced likelihood of continued breastfeeding to 12 months in a US [40] and Norwegian [52] study.

Even when available, not all women will be eligible for maternity leave, or for financial reasons able to take maternity leave. Therefore the level and type of support provided within their place of employment on return to work may influence their decision to breastfeed and duration of breastfeeding [53,54]. Employers should be encouraged, or even required, to offer supportive work arrangements to enable a woman to continue breastfeeding after she returns to work. This includes at the very least flexible scheduling and sufficient time to express milk and a designated lactation space other than a bathroom (toilet). Interestingly, a US study revealed that women working within the service and production/transportation industries—those who for financial reasons are most likely to return to work when their child is only months old—received a lower level of workplace support than women within the professional/ management industry [53].

In-hospital formula supplementation has been consistently shown to reduce the likelihood of a woman being able to attain exclusive breastfeeding by hospital discharge [55] and to have a detrimental effect on overall-duration of breastfeeding [56,57,58]. This study demonstrates however that the detrimental effect of formula is not just contained to the in-hospital use of formula but that the introduction of formula at any age has a negative effect on overall duration. Even the introduction of formula after the age of 6 months, by which time breastfeeding has been successfully established, was independently negatively associated with continued breastfeeding to 12 and 24 months. The reasons for giving formula will vary according to the age of the child, with perceived insufficient milk supply and poor latch being reasons for in-hospital use of formula [57], whereas mothers who first give formula after 6 months may do so for employment-related reasons or the perception that their infant was hungry [59]. Whatever the reason, women should be supported to overcome problems and circumstances which result in the introduction of formula in order to extend the duration of breastfeeding.

A limitation of this secondary analysis is that the primary purpose of the SMILE study was not to investigate continued breastfeeding practices. Therefore, we were unable to investigate important determinants such as a mother’s breastfeeding intentions and attitudes towards continued breastfeeding, her knowledge of recommendations related to breastfeeding duration and events which may have impacted on her ability to continue to breastfeed to 12 months and beyond. Exploration of these factors is warranted in future studies specifically designed to investigate continued breastfeeding practices amongst Australian women. A further limitation is that breastfeeding practices were self-reported and may therefore have been susceptible to social desirability bias and misreporting, although as the primary focus of this study was oral health this bias may not be as much of an issue compared to a study which focused on infant feeding practices. The findings with regards to country of birth may not be generalizable as recruitment was limited to those women who were sufficiently competent in English to complete the surveys, although the ethnic profile of the analysis population was similar to that of women who gave birth in South Australia in 2013 [30]. According to the most recent population census data for 2016, 36% of Australian women aged 20–40 years have completed a university degree [60], indicating that the study population was more highly educated than the general population. However, the census data does not account for women in this age group who may be currently studying at a university or who had dropped out of university. Despite the sociodemographic difference between participants and non-participants, a strength of the study was the intentional oversampling of participants from socially disadvantaged areas [25] which resulted in a socio-economically diverse analysis population which was relatively representative of the population from which it was drawn [30].

## 5. Conclusions

This study found that just under one-third of Australian women breastfed to 12 months as recommended in the Australian Infant Feeding Guidelines and fewer than 1 in 10 breastfed to 24 months as recommended by the World Health Organization. While there are benefits to be gained from breastfeeding of any duration, the majority of Australian infants and their mothers are missing out on the additional benefits of continued breastfeeding. The majority of factors associated with the practice of continued breastfeeding are potentially modifiable and could be used to identify those women who might benefit from additional breastfeeding support from health professionals in the hospital, their family and partners at home, and from employers in the workplace.

## Figures and Tables

**Table 1 ijerph-16-03980-t001:** Characteristics of SMILE participant mothers and children with known breastfeeding duration up to 24 months.

Characteristic	Total (n = 1450)		BF at 12 Months (n = 461)		BF at 24 Months (n = 108)	
	n	%	n	%	n	%
*Maternal Characteristics*						
Age (years)						
<25	190	13.1	28	6.1	9	8.3
25−34	958	66.1	315	68.3	61	56.5
≥35	298	20.6	118	25.6	38	35.2
Missing	4	0.3				
Highest level of education						
High school/vocational	707	48.8	141	30.6	38	35.2
Some ^a^ university or graduate	737	50.8	317	68.8	69	63.9
Missing	6	0.4	3	0.7	1	0.9
IRSAD ^b^ deciles						
IRSAD deciles 1–2	277	19.1	64	13.9	15	13.9
IRSAD deciles 3–4	301	20.8	82	17.8	21	19.4
IRSAD deciles 5−6	285	19.7	96	20.8	22	20.4
IRSAD deciles 7−8	269	18.7	84	18.2	27	25.0
IRSAD deciles 9−10	307	21.3	132	28.6	22	20.4
Missing	11	0.8	3	0.7	1	0.9
Country of birth						
Australian & New Zealand	1035	71.4	306	66.4	61	56.5
UK/Ireland	52	3.6	15	3.3	5	4.6
China	110	7.6	36	7.8	10	9.3
India	50	3.4	24	5.2	4	3.7
Asia-other	104	7.2	39	8.5	16	14.8
Other	89	6.1	38	8.2	11	10.2
Missing	10	0.7	3	0.7	1	0.9
Age of infant when returned to work						
≤12 months	659	45.4	210	45.6	34	31.5
Not by 12 months	512	35.3	205	44.5	65	60.2
Missing	279	19.2	46	10.0	9	8.3
Pre-pregnancy BMI (kg/m^2^)						
<25	762	52.6	301	29.5	68	63.0
≥25	586	40.4	136	14.3	29	26.9
Missing	102	7.0	24	5.2	11	10.2
Parity						
Primiparous	666	45.9	198	43.0	47	43.5
Multiparous	715	49.3	250	54.2	60	55.6
Missing	69	4.8	13	2.8	1	0.9
Partner’s feeding preference						
Prefers breastfeeding	934	64.4	364	76.8	82	75.9
Prefers bottle feeding or ambivalent	495	34.1	99	21.5	22	20.4
Missing	21	1.4	8	1.7	4	3.7
*Child characteristics*						
Sex						
Male	765	52.8	247	53.6	56	51.9
Female	685	47.2	214	46.4	52	48.1
Birth weight (g)						
<2500	94	6.5	23	5.4	6	5.6
2500–3999	1189	82.0	385	83.5	88	81.5
≥4000	155	10.7	47	10.2	13	12.0
Missing	12	0.8	4	0.9	1	0.9
Age received complementary foods						
<17 weeks	368	25.4	66	14.3	16	14.8
17−25 weeks	876	60.4	319	69.2	70	64.8
≥26 weeks	117	8.1	61	13.2	17	15.7
Missing	89	6.1	15	3.3	5	4.6
Age received formula						
<4 weeks	643	44.3	113	24.5	33	30.6
4−7 weeks	138	9.5	18	3.9	4	3.7
8−15 weeks	157	10.8	33	7.2	5	4.6
16−25 weeks	115	7.9	32	6.9	12	11.1
≥26 weeks	124	8.6	60	13.0	5	4.6
Never received formula	192	13.2	164	35.6	40	37.0
Missing	81	5.6	41	8.9	9	8.3

^a^ Commenced but did not complete University. ^b^ IRSAD: Index of Relative Socio-Economic Advantage and Disadvantage with decile 1 = most disadvantaged and 10 = most advantaged BF = breastfeeding; BMI= Body Mass Index.

**Table 2 ijerph-16-03980-t002:** Prevalence of breastfeeding up to 24 months (n = 1450).

Months	%	95% CI
Birth	94.9	93.8−96.0
1	82.2	80.2−84.2
3	66.6	64.2−69.0
6	51.2	48.6−53.8
12	31.8	29.4−34.2
18	12.1	10.4−13.8
24	7.5	6.1−8.9

**Table 3 ijerph-16-03980-t003:** Median duration of breastfeeding by mother’s country of birth.

Country	Median Duration (weeks)
Total	28.7
Australia/ New Zealand	26.5 ^a,b,c,d^
UK/Ireland	24.3
India	34.3 ^a^
China	34.8 ^b^
Asia-Other	34.8 ^c^
Other	42.5 ^d^

Kruskal-Wallis H = 26.441, df = 5, *p* < 0.001. Median duration was significantly different (p < 0.001) for Australian born mothers and groups with similar superscript letters e.g., ^a^ indicates that Australia was significantly different to India. ^b^ indicates that Australia was significantly different to China. ^C^ indicates that Australia was significantly different to Asia-Other country. ^d^ indicates that Australia was significantly different to Other country.

**Table 4 ijerph-16-03980-t004:** Factors independently associated with breastfeeding at 12 months and 24 months postpartum.

Characteristics	BF at 12 Months		BF at 24 Months	
	AOR	95%CI	AOR	95%CI
*Maternal characteristics*				
Age (years)				
<25	0.56	0.27−1.18	0.55	0.17−1.82
25−34	0.86	0.58−1.30	0.66	0.37−1.18
≥35	1.00		1.00	
Highest level of education				
High school/vocational	1.00		1.00	
Some ^a^ university or graduate	2.28	1.57−3.31	1.34	0.74−2.43
Country of birth				
Australian & New Zealand	1.00		1.00	
UK/Ireland	0.65	0.25−1.70	1.93	0.64−5.85
India	0.83	0.46−1.50	1.83	0.80−4.15
China	1.59	0.74−3.40	1.48	0.40−5.50
Other Asia	0.94	0.51−1.73	2.88	1.35−6.11
Other	1.55	0.78−3.07	1.91	0.81−4.51
Age of infant when returned to work				
By 12 months	1.00		1.00	
Not by 12 months	1.45	1.04−2.02	2.58	1.56−4.31
Pre-pregnancy BMI (kg/m^2^)				
<25	1.00		1.00	
≥25	0.56	0.40−0.80	0.66	0.37−1.16
Parity				
Primiparous	1.00		1.00	
Multiparous	1.52	1.08−2.15	1.01	0.60−1.70
Partner’s feeding preference				
Prefers breastfeeding	1.76	1.22−2.56	1.56	0.84−2.87
Prefers bottle-feeding or ambivalent	1.00		1.00	
*Child characteristics*				
Age received complementary foods				
Before 17 weeks	0.43	0.23−0.80	0.73	0.29−1.82
Between 17 and 25 weeks	0.68	0.40−1.15	0.91	0.44−1.79
At 26 weeks or later	1.00		1.00	
Age received formula				
Before 4 weeks	0.05	0.03−0.09	0.28	0.15−0.52
Between 4 and 7 weeks	0.03	0.02−0.07	0.10	0.02−0.44
Between 8 and 15 weeks	0.05	0.03−0.10	0.17	0.06−0.51
Between 16 and 25 weeks	0.05	0.03−0.11	0.47	0.21−1.07
At 26 weeks or later	0.15	0.08−0.28	0.21	0.08−0.58
Never received formula	1.00		1.00	

^a^ Commenced but did not complete University. AOR Adjusted Odds ratio, BF = breastfeeding, BMI = Body Mass Index

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
