# Peer review of "Determinants of Continued Breastfeeding at 12 and 24 Months: Results of an Australian Cohort Study"

_ijerph, 2019, doi:10.3390/ijerph16203980_

Round 1

Reviewer 1 Report

See attached file

Author Response

Thank you for your positive support of this paper and your constructive comments which we have addressed below.

Minor comments:

Throughout the article there is reference to the variable of partners “preferring breastfeeding”, which does not sound correct. Perhaps this is a language/translation issue which has been adopted from a previous publication, however I think this would be better reworded as “supporting breastfeeding”. Introduction:

Response: We have not changed the wording as women were specifically asked if their partner had a preference for how she should feed their baby. The options were ‘prefers breastfeeding’, ‘prefers formula feeding’, ‘no preference’ or ‘don’t know’; with the latter two being coded as ambivalent. While this is used as a proxy measure of breastfeeding support we did not specifically ask if the partner supported breastfeeding. Therefore it is more appropriate to describe this as partner’s method of feeding preference.

The first paragraph would benefit from an opening sentence or two outlining the WHO recommendations about exclusive breastfeeding for 6 months. This will provide context and lead into the current paragraph discussing breastfeeding beyond 12 months.

Response: The first paragraph now begins with the WHO recommendations regarding exclusive and continued breastfeeding.

Line 73 Recruitment: Do you have any estimation of the numbers of participants eligible for recruitment versus the number who participated?

Response: We have chosen not to report the number of eligible women, as all eligible women were not necessarily invited to participate.  For instance, for logistical reasons, it was not possible to visit each hospital every day.  As a result, women who say delivered late on a Friday may have been discharged before the research assistant visited on a Monday adn therfore did not haev the opportunity to participate.  Therefore, the difference between the number of participants and the number of eligible women cannot be used to calculate a response rate which usually reflects the number of women who declined to participate.

Statistical analysis:

Did you test for normality? It is not obvious whether the data was highly skewed or not

Response: No normality checks were undertaken as logistic regression analysis does not require that data be normally distributed. The outcome variables have binary outcomes (BF at 12 months Yes or No, BF at 24 months Yes or No) and the explanatory factors are categorical. Therefore, data are not expected to be normally distributed.

Line 91: “These variables were derived from questions on infant feeding practices….”. It is not clear throughout the manuscript whether/if you were able to distinguish between direct breastfeeding (at the breast) or an infant being fed expressed breastfeeding via a bottle. Please mention here.

 Response: While women were asked if they had ever or were currently expressing breast milk, no attempt was made to quantify the proportion of breast milk received either directly or indirectly or to differentiate between women who were directly or indirectly breastfeeding. The following sentence has been added. 

With regards to breastfeeding practices, no distinction was made between whether a mother was direct breastfeeding (at the breast) or feeding an infant expressed breast milk via a bottle.

Line 109: One of the age ranges for formula introduction is 4-25 weeks, which is a very broad duration. Using a shorter duration category could have considerably affected the results. Please clarify here why this broad range was chosen, or else mention in the discussion/as a limitation.

Response: We reran the analysis with additional smaller age ranges up to 26 weeks. The results were not substantially changed with all significant predictors remaining statistically significant and little or no change to the effect size. The new analysis is reflected in Table 4 and the Supplementary table 2 - Sensitivity analysis.

Line 111: “Those variables in the simple analysis with a p value < 0.20 were entered into a multivariable model” – could you please explain why this cut off was chosen – is this a standard statistical approach used by your research group?

Response: This is a standard procedure used by our group and recommended by other researchers to reduce the risk of residual confounding. This has been explained in the methods.

Maldonado, G. and S. Greenland (1993). "Simulation study of confounder-selection strategies." Am J Epidemiol 138(11): 923-936.

Results: · Subject characteristics: Please clarify if the demographic characteristics of the 1431 participants whose data is used in this paper differed to the 2112 who were completed a baseline questionnaire?

Response: The characteristics of the 1450 with known breastfeeding status up to 24 months were compared with the 662 subject for which we had baseline data but no data on BF status at 24 months.  Unsurprisingly, those that had dropped out were younger, less educated, more socially disadvantaged and more likely to have been born outside of Australia.  This is common in studies of this kind and was anticipated. Therefore we over recruited from the hospital that services the most disadvantaged women so that even after this non-random drop-out we had a socio-economically diverse analysis population which was relatively representative of the population from which it was drawn. 

We have acknowledged the socio-demographic differences between participants and non-participants in the results section and added a supplementary table comparing the characteristics of these two groups.

Lines 149: “The overall median duration of breastfeeding was 28.7 weeks….”: did you measure duration of exclusive breastfeeding?

Response: As the focus of this paper was on the duration of continued (any) breastfeeding, we have chosen not to present results related to the duration of exclusive breastfeeding and other feeding practices  which will be the subject of another publication.

Discussion

Line 244: “superiority of their cultural practices”: Please consider using a different word instead of “superiority”. Cultural practices are situation specific and influenced by many external factors – using the word “superiority” sounds a little patronising.

Response: It was not our intention to be patronising, nor do we think that this statement is patronising. Nevertheless we have revised this sentence for want of a synonym that the reviewer might find less patronising.

Line 246: how was pre-pregnancy BMI measured? Please clarify if this was self-reported.

Response: BMI was calculated based on self-reported weight and height.  This has been clarified in the methods.

Paragraph about women with higher BMI and lower breastfeeding rates: Also worth mentioning that breastfeeding should be encouraged as a means to reduce post-partum weight retention.

Response: Thank you for the suggestion but we  have chosen not to mention the association between breastfeeding and post-partum weight reduction as this association has not been consistently found.

Neville, C. E., M. C. McKinley, V. A. Holmes, D. Spence and J. V. Woodside (2014). "The relationship between breastfeeding and postpartum weight change—a systematic review and critical evaluation." International Journal of Obesity 38(4): 577-590.

Line 270 Paragraph about women returning to work: please acknowledge here the role of expressed breast milk feeding, which is very common in the USA due to short maternity leave policies. Does this play a role in extended breastfeeding?

Response: I am sorry but I have found it difficult to weave an acknowledgement of expressed breast milk feeding into this section.  As we did not consider breast milk expression in our analysis we are unable to answer the question as to whether breast milk expression extends breastfeeding. We already acknowledge that supportive environments which include provision for breast milk expression are necessary to enable a woman to continue breastfeeding after she returns to work.

Line 290: “complementary feeding”: complementary feeding usually means the introduction of solid food – please clarify as the paragraph is concerned with supplementation with infant formula.

Response: To avoid confusion the first sentence of this paragraph has been removed.

Conclusion paragraph

Whilst I agree with the sentiment in the conclusion about the need to support continued breastfeeding, surely the greatest public health and nutritional benefit would be derived from encouraging those who cease breastfeeding at 2-3 months to continue to 6 months. Although I realise this is not the topic of the paper, some concession should be made to the fact.

Response: We wholeheartedly agree with this comment.  However, as you have noted this was not the topic of the paper.  We have modified the conclusion slightly to acknowledge that there are benefits to be gained from breastfeeding of any duration.

Reviewer 2 Report

Thank you for your manuscript which adds to the debate about optimal breastfeeding. I have a fw queries and comments:

Introduction: you do not mention the known immunological and growth benefits of BM, Andreas et al 2015 has a good overview Methods: you exclude women who do not speak English. This is a potential source of bias as these women could be a particularly vulnerable group Results: How does the 50% of women educated to university level compare with the population of Adelaide, same with the ethnic mix? Did you look at BF in women with preterm infants on the NICU and whether that was a factor in BF or not? You do not mentions twins specifically, was there a difference between twins and singletons? Discussion: what are the BF facilities like in Adelaide, BF rooms in shopping centres etc. Day care: this can also be a limiting factor for feeding as women are often not allowed to BF on demand.

Author Response

Thank you for your constructive comments which we have addressed below.

Introduction: you do not mention the known immunological and growth benefits of BM, Andreas et al 2015 has a good overview.

Response: It was assumed that the short-term benefits of breastfeeding were well known and the focus of the introduction therefore was on the benefits of continued breastfeeding beyond 12 months and not the well-established immunological and growth benefits of BM in the first 12 months. We have revised the introduction to acknowledge the importance of breastfeeding and immunological properties of breast milk in infancy (i.e. 1st 12 months of life).

Methods: you exclude women who do not speak English. This is a potential source of bias as these women could be a particularly vulnerable group Results:

Response: We have acknowledged this as a limitation of the study.

How does the 50% of women educated to university level compare with the population of Adelaide, same with the ethnic mix?

Response: We have addressed this in the limitations.  A new supplementary table compares the characteristic of participants and non-participants as well as the characteristics of women giving birth in South Australia in 2013.  The ethnic mix of our sample was very close to the South Australian birth population. It should be noted that most migrants settle in major cities and not rural areas and therefore the South Australian figures are relevant to Adelaide.

Did you look at BF in women with preterm infants on the NICU and whether that was a factor in BF or not?

Response: We have clarified that although there were no exclusions on the basis of birth weight and gestational age, that women with infants in the NICU were not invited to participate. This was a requirement of ethics approvable from the hospitals. (Line 83)

You do not mentions twins specifically, was there a difference between twins and singletons?

Response: Although 34 sets of twins were recruited only 2 sets were include in the analysis population, therefore we could not investigate the statistical difference between singleton and twins.

Discussion: what are the BF facilities like in Adelaide, BF rooms in shopping centres etc. Day care: this can also be a limiting factor for feeding as women are often not allowed to BF on demand. 

Response: While we agree that lack of BF facilities may be a limiting factors we did not collect this information so cannot make any associations between availability of facilities and continued BF. We have acknowledged in our limitations that as the primary purpose of the SMILE study was not to investigate continued breastfeeding practices, we were unable to collect data on a variety of other factors which may impact on a woman’s breastfeeding practices and success.